# Polyphenols, L-Ascorbic Acid, and Antioxidant Activity in Wines from Rose Fruits (*Rosa rugosa*)

**DOI:** 10.3390/molecules26092561

**Published:** 2021-04-28

**Authors:** Andrzej Cendrowski, Marcin Królak, Stanisław Kalisz

**Affiliations:** Division of Fruit, Vegetable and Cereal Technology, Department of Food Technology and Assessment, Institute of Food Sciences, Warsaw University of Life Sciences-SGGW, Nowoursynowska 159C Str., 02-776 Warsaw, Poland; krolak93@gmail.com (M.K.); stanislaw_kalisz@sggw.edu.pl (S.K.)

**Keywords:** *Rosa rugosa*, fruit wines, fermentation process, polyphenols, L-ascorbic acid, antioxidants

## Abstract

The aim of the present study was to determine the influence of the winemaking process on the antioxidant potential and content of phenolic compounds and L-ascorbic acid in wines from the fruits of *Rosa rugosa*. The results obtained in this study clearly indicate that the fruits of the *Rosa rugosa* are a desirable raw material for the production of fruit wine. The parameters of the technological process of producing wines from rose fruits had a diversified influence on the tested quality characteristics. Aged wines contained phenolics levels of 473–958 mg/100 mL GAE. The final concentrations of ascorbic acid ranged from 61 to 155 mg/100 mL for the different variants of the wine. Wines revealed high antioxidant activity in assay with DPPH. On the basis of the obtained results, it can be assumed that all the applied variants of the winemaking process are suitable for rose fruit wine. Each variant ensured at least the stability of the antioxidant capacity.

## 1. Introduction

Antioxidant content in fruits as well as its associated antioxidant activity can be affected by physiological factors such as maturity and also by technological factors such as processing and storage conditions [1,2].

Among fruit products, wines are characterized by high antioxidant potential [3]. Currently, owing to the fact that wines have become an integral part of the culture of many countries, they are produced not only from grapes but also from many other fruits, including mangos and bananas [4,5]. Wines contain many classes of compounds, including phenolic compounds, which are the main components of wine’s antioxidant properties and significantly influence the sensory properties of wines such as mouthfeel, color, flavor, bitterness, and astringency [6,7,8]. The content of phenolic compounds in wines is influenced by many factors such as sulfur dioxide, growing conditions, geographic location and climatic conditions, production technology, pesticide residues and metal content in the must, and the aging process, including in-barrel and in-bottle [9,10,11,12]. SO_2_ (sulfur dioxide) in particular plays the key role in antioxidant capacity in commercial winemaking [13,14].

In Poland, the most popular fruits for wine production are the antioxidant-rich berry fruits, i.e., raspberries, strawberries, and currants [15]. Rose fruits deserve special attention as a potential source of natural antioxidants and other bioactive substances [16,17,18,19,20,21,22].

The results of a study by Meng et al. [23] and by Czyzowska et al. [24] show that wines made from Cherokee rose (*Rosa laevigata* Michx.), *Rosa rugose*, and *Rosa canina* have higher levels of total phenolics (TPC) than spine grape wines and Cabernet Sauvignon wines. The content of bioactive compounds in wine is influenced by winemaking technology [15,25]. The processes of fermentation and maceration that the fruit undergoes during wine production as well as the storage of wine decrease the content of polyphenolic compounds. Polyphenols are oxidized by reactive oxygen species, first to semiquinones and then to quinones. The rate of this reaction largely depends on pH: An increase in wine pH of one unit causes an increase in the concentration of phenolate ions, i.e., the acceleration of the oxidation of polyphenols [15,26]. In addition, the presence and arrangement of OH and OCH_3_ groups influence the antioxidant activity of polyphenols [27].

In theory, only one raw material is needed to produce wine: namely, grapes. However, in practice, in order to improve the quality, speed up the process, and increase the safety of production, one can also use: yeast, SO_2_, organic acids, calcium carbonate, lactic acid bacteria, nutrients, clarifying agents, sugar, and alcohol. To make rose fruit wine, one needs four basic raw materials: rose hips, sugar, water, and yeast. Among the additives, enzyme preparations and dyes can also be used. Simply put, the start of wine production is to mix the above-mentioned ingredients (with the exception of clarifiers and dyes, which are added after the fermentation process) in the right proportions. The final quality of the wine depends on factors such as the type of grape strain or strains used, the fruit species (single or multiple fruits), the proportion and sequence of adding ingredients, the use of enzyme preparations, the length of the maceration of the pulp before pressing, fermentation in must or pulp, type of yeast strain, form of yeast used, the type of fermentation, and the fermenter used [28,29,30]. The most important stage in wine production is fermentation. The purpose of fermentation is to convert sugar into ethanol in such a way that the least undesirable by-products are formed, and the natural aromas and flavors of the fruit are preserved as much as possible. Depending on the conditions, it may last from several days to several weeks. The process of converting sugar into alcohol includes the entire glycolysis process, wherein the sugar is converted to pyruvate. The yeast then converts the pyruvate into acetaldehyde with the release of CO_2_. The final product of yeast metabolism, i.e., ethanol, is formed from the aldehyde [31]. The initial addition of yeast is important for the proper fermentation process. It can be added as dry active yeast or as mother yeast. Depending on the yeast strain used and the intended nature of the wine, the optimal fermentation temperature may vary but is most often between 20 and 30 °C for red wines and 10 to 17 °C for white wines. For fruit wines, the temperature is between 10 and 30 °C depending on the raw material used and the final character of the wine. The fermentation is completed when the desired alcohol or residual sugar content is reached. After the fermentation process is completed, the wine undergoes further technological processes such as stabilization, clarification, maturation (aging), coupage, and bottling [28,29].

In Poland, roses from the species *Rosa rugosa* are grown on commercial plantations. Despite the many years of tradition of using rose fruit and the enormous technological potential of this material, there is little literature data describing the use of this material in the production of fruit wine.

Hence, the aim of the study was to determine the influence of the winemaking process on the antioxidant potential and content of phenolic compounds and L-ascorbic acid in wines from fruits of *Rosa rugosa*.

## 2. Results and Discussion

The study investigated the influence of the technological process on the content of bioactive compounds in wines from fruits of *Rosa rugosa*. The factors differentiating the technological process of wine production were different proportions of fruit, enzymatic treatment or its absence, and fermentation in must or pulp.

### 2.1. Physicochemical Characteristics of the Varieties of Rosa rugosa Wines Produced

Values of the physicochemical parameters analyzed in *Rosa rugosa* wines after 3 months of aging are shown in Table 1.

By fermenting the musts, higher alcohol contents could be achieved. Variant 1 had at least 12% more alcohol than variants 3, 4, 5, and 6. In the case of musts, the proportion of fruit had an impact on the final alcohol content. A higher proportion of fruit resulted in a lower alcohol content. In this case, variant 2 had over 5% less alcohol than variant 1. Interestingly, in the case of pulp fermentation, although the fruit content significantly influenced the fermentation kinetics, it did not affect the final alcohol content. There appears to be some specific correlation between the substances from the *Rosa rugosa* fruits that increases yeast stress while being tolerant to ethanol [32].

The wine variants that were subjected to the enzymatic treatment differed in their pH value by no more than 3%, which had no effect on the technological process. In addition, variants 3 and 5 had a clearly higher pH than the others. Variant 3 had over a 6% higher pH value than variant 4, and variant 5 had over 10% higher content than variant 6. Most likely, this could have been caused by pectin degradation by pectinolytic enzymes and damage to the fruit cell, which increased the amount of must obtained and accelerated extraction of substances contained in the cell, including acids.

The titratable acidity of the tested wine variants was in the range of 10.14–14.34 g/100 mL. Especially large differences were seen between wine variant 3 and 6. This was probably due to some unpredictability in the fermentation process. Wines obtained from the same grape varieties from the same vineyard and by the same technological process may show even greater differences in the value of titratable acidity than between the rose hip wines. Differences in titratable acidity can be as high as 20% [33].

Volatile acidity is the sum of all volatile acids contained in the product and is not limited to acetic acid only. Volatile acids include, for example, short-chain carboxylic acids from formic acid with 1 C atom to caprylic acid with 7 carbon atoms [34]. Volatile acidity is an important parameter indicating the correctness of the fermentation process carried out. If air enters in the initial stage of fermentation when the alcohol content is only a small percent, contamination with acetic acid fermentation bacteria may occur, which will adversely affect the quality of the wine [28,29]. In the case of the pulp-fermented variants, the higher the proportion of fruit, the higher the volatile acidity value. Variants 5 and 6 had over 37% higher acetic acid content than variants 3 and 4. On the other hand, fermentation in must allowed for obtaining a lower value of volatile acidity than in the pulp, which was visible in wines with 50% fruit content. Variant 2 showed approximately 30% lower acetic acid content than variants 5 and 6. The sugar content in the studied variants of *Rosa rugosa* wines was characteristic of semi-sweet wines described by other authors [35].

### 2.2. Total Phenolic Content

The Folin–Ciocalteu method is usually used for determination of total polyphenol content, but the reagent is nonspecific. The results of the determinations made by this method are affected by the presence of reducing sugars, aromatic amines, sulphur dioxide, ascorbic acid, organic acids, and other compounds, making the results often overstated. According to Prior and others and Huang and others [36,37] this method is based on SET-transfer of a single electron, and can thus be considered to be one of the methods for determination of antioxidant activity. This was also confirmed by the studies of other authors, who observed a high regression coefficient between the amounts of substances that reacted with the Folin–Ciocalteu reagent and antioxidant activity determined by a DPPH test [38,39].

The differences between our values and the published results could be primarily affected by the nature of the analyzed wines, i.e., by their actual contents of phenolic compounds. It is known that many internal and external factors significantly influence the concentration of phenolic compounds in wines. The total phenolic content of the wines in our study was found to be in range of 538–951 and 556–958 mg/100 mL GAE for aged and young wines, respectively (Figure 1). The content of total phenolic compound in our wines was several times higher than that of *Rosa rugosa, Rosa canina*, and *Rosa laevigata* wines made by Czyżewska et al. and Meng et al. [23,24]. For variants 1, 3, 5, and 6 of our wines, high stability of phenolic compounds during the technological process was demonstrated (Figure 1). Changes in polyphenol content were not greater than 10%.

Only variants 2 and 4 showed about 20% loss of polyphenols. Laskowska et al. [40] also obtained very good polyphenol stability. In the case of their wine obtained from rose fruit, the change in polyphenol content did not exceed 6%. On the other hand, the study by Czyżowska et al. [24] showed more than 60% loss of phenolic compounds during the production of wine from rose fruit. It seems that the stability of polyphenols in wines of rose fruit is very complex. Minimal differences, which are difficult to control, may determine the loss of phenolic compounds in wines from rose fruits. In the case of the final product, i.e., wine, after 3 months of aging, the statistical analysis showed that the content of phenolic compounds was only affected by the share of fruit (Table 2).

The higher the fruit content, the higher the total polyphenol content. In the final product, the fermentation environment and the enzymatic treatment, or absence thereof, did not directly affect the total polyphenol content. However, statistical analysis of the correlation between the fruit share and the fermentation environment showed its indirect influence on the polyphenol content of *Rosa rugose* wines obtained. Fermentation in must counteracted the impact of fruit content. Statistical analysis showed no significant differences in the content of phenolic compounds between variants 1 and 2 of the wine. Correlations between the proportion of fruits and the use of enzyme treatments were also investigated. Here, too, the mediating effect of the enzymatic treatment or its absence was demonstrated. In the case of wines with 50% fruit content, the enzymatic treatment increased the content of total polyphenols with statistical significance. The wine production process allowed for good retention of phenolic compounds. Two-thirds of wine variants showed very good stability of phenolic compounds, and only 1/3 showed losses below 23%. Even variant 4 of the wine, with the lowest total polyphenol content, showed 473 mg/100 mL, which is more than fresh cherries (460 mg/100 g). Fresh cherries are considered a rich source of these compounds [41]. On the other hand, variant 6 of the wine contained twice as many polyphenols as fresh cherries.

### 2.3. L-Ascorbic Acid Content

After the end of the fermentation process, a decrease in the L-ascorbic acid content was detected in all wine variants (Figure 2).

For variant 1, the decrease compared to the L-ascorbic acid content before fermentation was about 31%. Similarly, in variant 2, it was approximately 36%; variant 3, approximately 26%; variant 4, approximately 42%; variant 5, approximately 26%; and variant 6, approximately 27%. According to other authors, the losses of L-ascorbic acid in the process of alcoholic fermentation also amounted to several percent [24,40]. Several factors may have influenced the loss of L-ascorbic acid. Laskowska et al. [40] used dried yeast in their experiment, and in this study, mother yeast was used. The use of dried yeast allowed for shortening of the time of yeast dominating the environment as compared to the yeast mother, which resulted in a shorter contact of L-ascorbic acid with dissolved oxygen [28,29,42]. Moreover, the use of the mixing process introduced additional oxygen, which could result in additional degradation of the vitamin. The raw material itself could be the reason for the differences in L-ascorbic acid losses. The content of nitrates in *Rose rugosa* fruits may vary from 40 to 150 mg per kg of fruit, depending on the cultivation methods and conditions [42]. One of the properties of L-ascorbic acid is that it prevents the formation of nitrosamines by reducing nitrates to nitric oxide. Perhaps the raw material used in this experiment had more nitrates than that used by Laskowska et al. [40]. In addition, it must be remembered that L-ascorbic acid is a secondary antioxidant and can regenerate primary antioxidants, which are polyphenols. Small losses of polyphenols in the tested wine variants could be caused by the protective effect of L-ascorbic acid, which could also cause additional losses of this vitamin [43]. L-ascorbic acid was lost in all wine variants during the aging process. Variant 6 showed the greatest stability of L-ascorbic acid. Statistical analysis showed that differences in the content of L-ascorbic acid before fermentation and in wine just after the fermentation process was completed were insignificant. After a month of aging, the losses of this vitamin amounted to approximately 30–40%. For the remaining months, the content of L-ascorbic acid in the 6th variant of the wine was stable. It is known that the chemical reactions in wine can be very slow. Calcium tartrate may crystallize from wine for months, causing its turbidity, and the alcohol maturation processes may take years [28,29,44]. In the case of the final product, i.e., wine after 3 months of aging, the statistical analysis showed that the L-ascorbic acid content was influenced by all three variables (Table 3).

A higher proportion of fruit resulted in a higher content of L-ascorbic acid. The must-fermented variants showed a higher L-ascorbic acid content than the pulp-fermented variants. On this basis, it could be concluded that fermentation in musts allowed for better L-ascorbic acid preservation during the technological process. The pulp-fermented variants with the enzymatic treatment applied showed statistically significantly lower L-ascorbic acid content than those without the enzymatic treatment. A statistical analysis of the correlation between the share of rose fruits and the fermentation environment was performed. The analysis showed that the proportion of fruit, in the case of fermentation in must, was less important for the preservation of L-ascorbic acid in the final product than the fermentation environment. In the case of fermentation in the pulp, the share of fruit was also important in determining the final content of this vitamin. The correlations between the proportion of fruit and the presence or absence of enzyme treatment were also statistically examined. There were no statistically significant differences between the variants. Based on this correlation, it could be concluded that the proportion of fruit and the enzymatic treatment or lack thereof were of secondary importance in shaping the L-ascorbic acid content in the finished wines. On the basis of the two correlations carried out, it could be concluded that the factor that had the greatest impact on the content of L-ascorbic acid in the final product was the fermentation environment, and wine production in musts allowed better L-ascorbic acid preservation than in the pulp. The decrease in the content of L-ascorbic acid for all variants of wine was on average approximately 63%, which meant for the worst variant (variant 4) 61 mg of this vitamin in 100 mL of wine. A significant amount of vitamin C is approximately 12 mg/100 g or 100 mL [45], and variant 4 had 5 times as much. Assuming that the daily requirement for vitamin C is about 80 mg/day/person, except for variant 4, 100 mL of the obtained wines satisfied the daily requirement for this vitamin, and 100 mL, in the case of wine variant 2, provided almost twice the dose.

### 2.4. Antioxidant Capacity

The radical scavenging activity of the *Rosa rugosa* wines was determined from their DPPH radical quenching ability. The DPPH test allows us to determine the ability to scavenge radicals through the HAT mechanism [36]. The relatively stable organic radical, DPPH, has been widely used in the determination of antioxidant activity of single compounds as well as of other different products [46]. Some studies showed that antioxidant activity of plant products was correlated with total phenolics rather than with individual phenolic compounds, so the total phenol content was investigated in this study [47]. It is important to state that different phenols develop different activities depending on their chemical structure (phenolic acids, flavanols, antocyanidins, stilbens), and the capacity for scavenging free radicals from these classes of compounds differs. The antioxidant properties of a single compound within a group can be different, and thus the same levels of phenolic compounds do not necessarily correspond to the same antioxidant responses [48].

The antioxidant capacity shows how effectively the tested variants of prepared wines, before and 3 months after fermentation, can scavenge free radicals formed in our bodies. The antioxidant capacity is often equated with polyphenol content. The high content of phenolic compounds results in a high free radical scavenging capacity. However, polyphenol compounds differ in their antioxidant activity. In addition, it must not be forgotten that synergistic interactions, such as, for example, between vitamin C and other phenols, or inhibitory interactions between the compounds can take place. Therefore, for example, raspberries, despite the fact that they have a lower polyphenol content than black currants, bind the DPPH radical better [49,50,51]. Figure 3 shows the IC50 values for the prepared variants of *Rosa rugosa* wine before fermentation and 3 months after fermentation.

Statistical analysis showed that the antioxidant capacity for variants 1 and 3 was stable. No statistically significant differences were found. The other variants showed an increase in antioxidant capacity. On this basis, it can be assumed that the antioxidant capacity of *Rosa rugose* wine may be stable during the technological process and may even increase. Although an increase in antioxidant capacity during technological processes is rare, it is possible. Alcoholic fermentation reduces the loss of antioxidant compounds. In addition, pectins, even without pectinolytic enzymes, are broken down during the alcoholic fermentation process. The process of wine maturation is a complex and still little-understood phenomenon in which hundreds of compounds, including polyphenols, participate. Similar to pectins, they can decompose into compounds of lower molecular weight (galacturonic acid), increasing the content of polyphenols [52]. Esters of phenolic acids also show better antioxidant properties than phenolic acids alone [53]. Polyphenols can also react with each other to increase antioxidant capacity, such as, for example, epigallocatechin gallate, which is a much stronger antioxidant than gallic acid or catechins [54]. We cannot forget about synergistic interactions, which in the fruit were limited by its structure and which probably occur in the obtained wines between individual compounds [49,50,51]. Such variation in antioxidant activity was also observed by Kallithraka, Salacha, and Tzourou [55] in wine. These authors believed that this may have resulted from the reactions between oxidized polyphenols and the formation of new compounds of antioxidant character. Therefore, predicting the antioxidant capacity of processed fruit was significantly impeded.

In the case of the final product, which was considered to be *Rosa rugosa* wine after 3 months of aging, the three-factor analysis of variance for the concentration of 0.625% showed that the DPPH radical scavenging capacity was influenced by all three variables (Table 4).

The greater the proportion of fruit, the greater the content of L-ascorbic acid and polyphenols, which translated into a better antioxidant capacity. The pulp-fermented wine variants had a better DPPH radical scavenging capacity. This suggested that the water-insoluble dry matter compounds and *Rosa rugosa* fruit had a positive effect on the antioxidant capacity of the wines obtained. This fact was confirmed by the obtained IC50 values, which showed that variants 5 and 6 had the best antioxidant capacity of the obtained fruit wines. Fruit wine variants after enzymatic treatment scavenged the DPPH radical worse than those without enzymatic treatment. This could be due to the initial greater losses of L-ascorbic acid and increased temperature during the enzymatic treatment.

On the basis of the obtained results, it could be assumed that all of the applied variants of obtaining the rose fruit wine were suitable because each one ensured at least the stability of the antioxidant capacity. Variants 5 and 6 proved to be the best with the lowest IC50 values, which speaks in favor of pulp fermentation when it comes to the production of rose fruit. Despite the large losses of L-ascorbic acid, wines show very good antioxidant properties, which may confirm the protective effect of L-ascorbic acid on phenolic compounds.

## 3. Materials and Methods

### 3.1. Chemicals and Reagents

Hydrochloric acid, anhydrous sodium carbonate, and Folin–Ciocalteu reagent were purchased from Chempur (Piekary Śląskie, Poland). Gallic acid anhydrous, 2,2-diphenyl-1-picrylhydrazyl (DPPH), L-ascorbic acid, oxalic acid, *m*-phosphoric acid, 6-hydroxy-2,5,7,8-tetramethylchromane-2-carboxylic acid (Trolox), and methanol were purchased from Sigma–Aldrich (Poznan, Poland). All reagents were of analytical grade.

### 3.2. Plant Material

Fresh fruits of *Rosa rugosa* were collected from plantation of the company “Polska Róża”, located in Kotlina Kłodzka (16°39′ E 50°27′ N, Poland). Soil fertilization and shrub cutting were performed in accordance with the cultivation recommendations for this species. The fruits were collected in October 2018 at the stage of collective maturity. The temperature during the vegetative period was close to the annual averages. There were no extremes that could affect normal development of *Rosa rugosa*. Seedlings of *Rosa rugosa* were planted into a loose, loam-type soil mixture. The soil was slightly moist with added manure. The pH of the soil was about 6–6.5. Before planting and during the first year after the planting, no mineral fertilizers were used. The raw material (20 kg) was stored until the beginning of the study at −18 °C. Before starting the research work, the rose fruits were subjected to pretreatment, which included washing under running water and manual dressing so as to separate damaged, overripe, and rotten fruits, unsuitable for further processing.

### 3.3. Winemaking Procedure

Six different wine variants were prepared from the fruits of the *Rosa rugosa*. The prepared variants differed in fruit content (35% or 50%), fermentation in must after enzymatic treatment, or fermentation in pulp without or after enzymatic treatment. The preparation of different wine variants is shown in Appendix A in Figure A1 and Table A1. Rohapect^®^ 10 L, a universal pectinolytic enzyme preparation (AKE Łaszkiewicz, Dzida, Poland), was used for the enzymatic processing of the raw material. The enzymatic treatment was carried out at the temperature of 50 °C in accordance with the manufacturer’s recommendation, Rohapect^®^ 10 L.

Mother yeast and universal wine broth (Spiritferm, Warsaw, Poland) containing phosphorus, nitrogen, and vitamin B1 compounds were used for fermentation. The yeast mother was prepared by the Department of Microbiology of the Faculty of Food Sciences of the Warsaw University of Life Sciences by inoculating the yeast *Saccharomyces cerevisiae* Steinberg in apple juice reconstituted from the Tymbark concentrate, Poland. Variants 1 and 2 of the wine were additionally sweetened with sugar to reduce the differences in the sugar content in the compared wine variants. However, variants 3, 4, 5 and 6 of the wine were fortified with rectified spirit containing 96.5% alcohol (Polmos, Bielsko-Biała, 

Poland) to reduce the differences in alcohol content. After completing the fermentation process, the wine was decanted from the sediment and aged in bottles at room temperature for a period of 3 months. The experiment was carried out in two technological duplicates.

### 3.4. Physicochemical Analysis

Alcohol: The determination was made with the use of a Tralles hydrometer, calibrated to measure the alcohol content from 10% to 20%. The measurement consisted in measuring the density of the distillate obtained from the wine. The result was given in degrees Tralles (% by volume).

pH: The determinations were made in accordance with the Polish Standard PN-EN-1132: 1999 using a pH meter [56].

Titratable acidity: The determinations were made in accordance with Polish Standard PN-EN-12147: 2000 expressed as malic acid [57].

Volatile acidity: The determination was made by direct distillation and the obtained distillate was titrated with 0.1 M sodium hydroxide solution against phenolphthalein. Volatile acidity was expressed as acetic acid.

Sugar: The content of total sugars was determined by the Luff–Schoorl method [58].

### 3.5. Determination of Total Phenolic Content

Total phenolic content (TPC) was determined by using Folin–Ciocalteu reagent according to the well-known method, with slight modifications [59].

In brief, samples of pulp, must, or wine were transferred to test tubes and centrifuged for 5 min at 5000 rpm. 1 mL of centrifuged pulp, must, or wine was diluted in 100 mL of distilled water. Next, 0.2 mL of the solution was mixed with 0.4 mL of Folin–Ciocalteu reagent, 4 mL of distilled water, and 2 mL of 15% sodium carbonate. After 1 h of blend incubation in a dark place at a temperature of 25 °C, the absorbance measurements were taken with a UV-Vis spectrophotometer at a wavelength of 765 nm (25 °C) against mixed reagents. Total phenolic content was expressed as mg gallic acid equivalents (GAE) per 100 mL based on the prepared calibration curve. The equation obtained from the calibration curve of gallic acid in the range of 5–50 mg/100 mL was y = 0.036x + 0.0999 (r = 0.9966). The test was performed in triplicate.

### 3.6. Determinations of L-Ascorbic Acid

Determination of L-ascorbic acid in the tested samples was performed by HPLC using the method previously stated in the literature with our own modification [60,61]. In order to protect L-ascorbic acid in the test sample, a 2% oxalic acid solution was used as the solvent. In brief, samples of pulp, must, or wine were transferred to test tubes and centrifuged for 10 min at 5000 rpm. 1 mL of centrifuged pulp, must, or wine was diluted in 10 mL of 2% oxalic acid. This solution was filtered through a 0.45 µm PTFE syringe filter to a chromatographic vial. The sample volume of 20 µl from each chromatographic vial was injected into the HPLC system with a UV-Vis detector (Shimadzu, Kyoto, Japan). The separation was carried out using an Onyx Monolithic C18, 100 × 4.6 mm column (Phenomenex, Torrance, CA, USA), at 25 °C. 0.1% *m*-phosphoric acid was used as eluent using a flow rate of 1.0 mL/min. The analysis was performed in an isocratic system. L-ascorbic acid was determined at 245 nm.

The equation obtained from the calibration curve of L-ascorbic acid in the range of 0.5–140 mg/100 mL was y = 345802x − 47799 (r = 0.9999). L-ascorbic acid was expressed in mg/100 mL sample. The test was performed in triplicate.

### 3.7. DPPH Assay

The DPPH (2,2-diphenyl-1-picrylhydrazyl) free radical scavenging activity was determined using the method stated in the literature [62]. The method used was to assess the color change of the methanolic DPPH radical solution from purple to light yellow. The DPPH radical is quenched by removing electrons from the antioxidant substance, which results in a color change registered spectrophotometrically [63].

To 2.9 mL of 0.1 mM methanolic solution of 1,1-diphenyl-2-picrylhydrazyl radical (DPPH), 0.1 mL of methanol extract of pulp, must, wines, or standards (trolox) of appropriate concentration (Appendix A, Table A2) was added. The absorbance (A_30_) was measured at a wavelength of λ = 517 nm in 1 cm cuvettes after 30 min incubation at room temperature against a reference sample (3 mL methanol). The control absorbance (2.9 mL of 0.1 mM methanolic DPPH solution and 0.1 mL of methanol) was measured at the beginning and end of the analysis. The arithmetic means of the absorbance for DPPH were the A_DPPH_ values.

The DPPH radical reduction ability (antioxidant activity, AA%) of the tested samples was calculated according to the formula:AA% = [(A_DPPH_ − A_30_)/A_DPPH_] × 100(1)
where A_30_—absorbance of the tested sample, A_DPPH_—control absorbance.

The IC50 was calculated from the equation describing the dependence of the degree of scavenging of the DPPH radical (%) on the concentration of the methanol extract/standard in the tested sample.

### 3.8. Statistical Analysis

All results are expressed as the mean ± standard deviation (SD) of three replicates. Statistical analysis was conducted using Statistica version 10.0 (StatSoft Poland, Cracow, Poland). Assessment of the effects of fruit concentration, type of fermentation (in must or pulp), and enzymatic treatment or its absence on the tested features was performed using a three-way ANOVA. Tukey’s test was used for the calculations.

## 4. Conclusions

Rose fruits are not routinely used in the wine industry, but their wide availability in Poland and the presence of desired compounds showing antioxidant and health benefits indicate that they are a great resource for use in industrial production.

The results obtained in this study clearly indicate that the fruits of the *Rosa rugosa* are a desirable raw material for the production of fruit wine. The parameters of the technological process of producing wines from rose fruits had a diversified influence on the tested quality characteristics. The variants fermented in must showed a higher content of polyphenols than those fermented in pulp. In the case of wines with 50% fruit content, the enzymatic treatment increased the content of total polyphenols with statistical significance. Overall, the wines obtained showed a very good stability of phenolic compounds. On the basis of the correlations carried out, it can be concluded that the factor that had the greatest impact on the content of L-ascorbic acid in the final product was the fermentation environment, and wine production in musts allowed better L-ascorbic acid preservation than in the pulp. On the basis of the obtained results, it can be assumed that all the applied variants of producing the rose fruit wine are suitable because each one ensured at least the stability of the antioxidant capacity. Variants 5 and 6 turned out to be the best with the lowest IC50 values, which indicated the use of pulp fermentation in the production of *Rosa rugosa* wine.

## Figures and Tables

**Figure 1 molecules-26-02561-f001:**
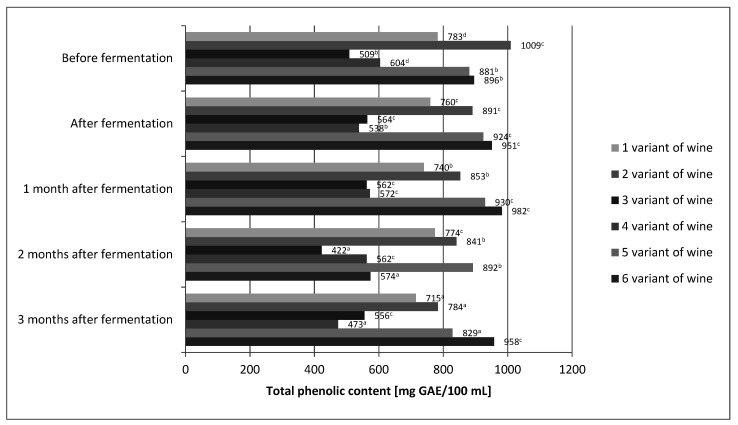
The concentration of polyphenols in different variants of wine prepared from fruits of *Rosa rugosa*. The values with different superscript letters are significantly different (*p* < 0.05).

**Figure 2 molecules-26-02561-f002:**
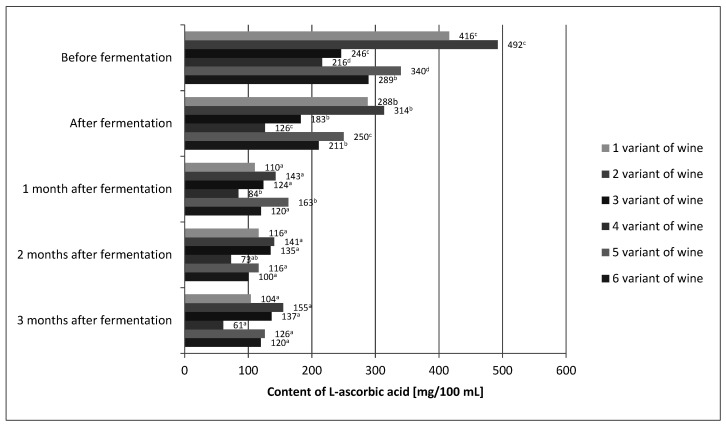
The content of L-ascorbic acid in variants of the prepared wine from fruits of *Rosa rugosa*. The values with different superscript letters are significantly different (*p* < 0.05).

**Figure 3 molecules-26-02561-f003:**
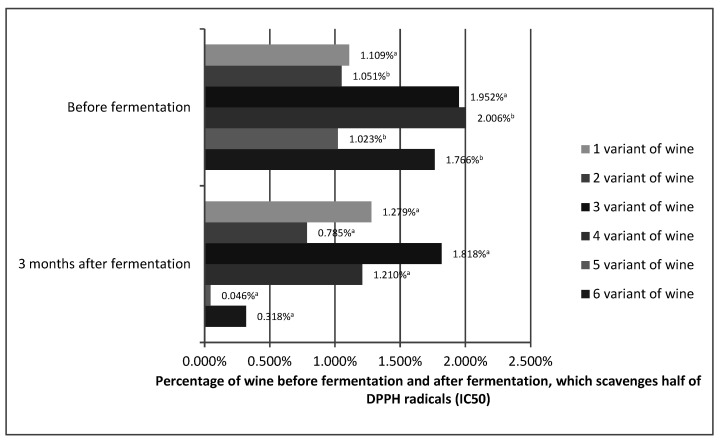
Percentage of wines from *Rosa rugose* fruits before fermentation and 3 months after fermentation, which scavenges half of the DPPH radicals (IC50). The values with different superscript letters are significantly different (*p* < 0.05).

**Table 1 molecules-26-02561-t001:** Mean and standard deviation of physicochemical parameters in different variants of wine prepared from fruits of *Rosa rugosa*.

Variant of Wine	Alcohol Content, %	pH	Titratable Acidity, g/100 mL	Volatile Acidity, g/L	Sugars, g/L
1	14.60 ± 0.07	3.65 ± 0.01	12.40 ± 0.02	1.278 ± 0.02	51.95 ± 1.23
2	13.76 ± 0.02	3.65 ± 0.01	14.16 ± 0.02	1.137 ± 0.01	51.87 ± 1.89
3	12.78 ± 0.06	3.63 ± 0.01	10.14 ± 0.01	1.167 ± 0.01	51.51 ± 1.56
4	12.26 ± 0.10	3.60 ± 0.01	10.50 ± 0.02	1.195 ± 0.01	51.08 ± 1.61
5	12.78 ± 0.07	3.68 ± 0.01	13.67 ± 0.01	1.609 ± 0.02	51.68 ± 1.42
6	12.27 ± 0.10	3.61 ± 0.01	14.34 ± 0.01	1.726 ± 0.01	51.32 ± 1.21

**Table 2 molecules-26-02561-t002:** Multivariate analysis of variance of the polyphenol content of wines after 3 months of aging on the share of fruit (a), fermentation in must or pulp (b), enzymatic treatment or its absence (c) and a combination of factors a * b and a * c.

% Share of Fruit		Average	Standard Error	Start of the Confidence Interval (95%)	End of Confidence Interval (95%)	Homogeneous Group
35		635	15.5	602	669	A
50		806	15.5	772	840	B
**Fermentation Environment**		**Average**	**Standard Error**	**Start of the Confidence Interval (95%)**	**End of Confidence Interval (95%)**	**Homogeneous Group**
in the pulp		704	11	680	728	A
in the must		738	19	696	779	A
**The Enzyme Treatment Used**		**Average**	**Standard Error**	**Start of the Confidence Interval (95%)**	**End of Confidence Interval (95%)**	**Homogeneous Group**
without		709	19	668	751	A
with		732	11	709	756	A
**% Share of Fruit**	**Fermentation Environment**	**Average**	**Standard Error**	**Start of the Confidence Interval (95%)**	**End of Confidence Interval (95%)**	**Homogeneous Group**
35	in the pulp	515	15.5	481	548	A
50	in the must	719	26.9	661	778	B
35	in the must	756	26.9	698	815	B
50	in the pulp	893	15.5	860	927	C
**% Share of Fruit**	**The Enzyme Treatment Used**	**Average**	**Standard Error**	**Start of the Confidence Interval (95%)**	**End of Confidence Interval (95%)**	**Homogeneous Group**
35	with	594	15.5	560	628	A
35	without	677	26.9	618	735	AB
50	without	742	26.9	683	801	B
50	with	871	15.5	837	904	C

The values with different capital letters (A–C) are significantly different (*p* < 0.05).

**Table 3 molecules-26-02561-t003:** Multivariate analysis of variance of the L-ascorbic acid content in wines after 3 months of aging on the share of fruit (a), fermentation in must or pulp (b), enzymatic treatment or its absence (c), and a combination of factors a * b and a * c.

% Share of Fruit		Average	Standard Error	Start of the Confidence Interval (95%)	End of Confidence Interval (95%)	Homogeneous Group
35		109	2.25	105	114	A
50		141	2.25	136	145	B
**Fermentation Environment**		**Average**	**Standard Error**	**Start of the Confidence Interval (95%)**	**End of Confidence Interval (95%)**	**Homogeneous Group**
in the pulp		111	1.59	107	114	A
in the must		139	2.75	133	145	B
**The Enzyme Treatment Used**		**Average**	**Standard Error**	**Start of the Confidence Interval (95%)**	**End of Confidence Interval (95%)**	**Homogeneous Group**
with		104	1.59	101	108	A
without		146	2.75	140	152	B
**% Share of Fruit**	**Fermentation Environment**	**Average**	**Standard Error**	**Start of the Confidence Interval (95%)**	**End of Confidence Interval (95%)**	**Homogeneous Group**
35	in the pulp	98.8	2.25	93.9	104	A
35	in the must	120.1	3.89	111.6	129	B
50	in the pulp	122.9	2.25	118	128	B
50	in the must	158.1	3.89	149.7	167	C
**% Share of Fruit**	**The Enzyme Treatment Used**	**Average**	**Standard Error**	**Start of the Confidence Interval (95%)**	**End of Confidence Interval (95%)**	**Homogeneous Group**
35	with	71.4	2.25	66.5	76.3	A
50	with	137.4	2.25	132.5	142.3	B
50	without	143.7	3.89	135.2	152.1	B
35	without	147.5	3.89	139	156	B

The values with different capital letters (A–C) are significantly different (*p* < 0.05).

**Table 4 molecules-26-02561-t004:** Multivariate analysis of variance of the dependence of DPPH radical scavenging ability by 0.625% wine solutions after 3 months of aging on the share of fruit (a), fermentation in must or pulp (b), enzymatic treatment or its absence (c), and a combination of factors a * b and a * c.

% Share of Fruit		Average	Standard Error	Start of the Confidence Interval (95%)	End of Confidence Interval (95%)	Homogeneous Group
35		26.1	0.54	25.0	27.3	A
50		61.1	0.54	59.9	62.3	B
**Fermentation Environment**		**Average**	**Standard Error**	**Start of the Confidence Interval (95%)**	**End of Confidence Interval (95%)**	**Homogeneous Group**
in the must		39.1	0.66	37.6	40.5	A
in the pulp		48.2	0.38	47.4	49.0	B
**The Enzyme Treatment Used**		**Average**	**Standard Error**	**Start of the Confidence Interval (95%)**	**End of Confidence Interval (95%)**	**Homogeneous Group**
with		40.3	0.38	39.5	41.2	A
without		46.9	0.66	45.5	48.4	B
**% Share of Fruit**	**Fermentation Environment**	**Average**	**Standard Error**	**Start of the Confidence Interval (95%)**	**End of Confidence Interval (95%)**	**Homogeneous Group**
35	in the pulp	23.9	0.54	22.7	25.0	A
35	in the must	28.4	0.94	26.4	30.4	B
50	in the must	49.7	0.94	47.7	51.7	C
50	in the pulp	72.5	0.54	71.3	73.7	D
**% Share of Fruit**	**The Enzyme Treatment Used**	**Average**	**Standard Error**	**Start of the Confidence Interval (95%)**	**End of Confidence Interval (95%)**	**Homogeneous Group**
35	without	23.6	0.94	21.6	25.7	A
35	with	28.6	0.54	27.5	29.8	B
50	with	52.0	0.54	50.8	53.2	C
50	without	70.2	0.94	68.2	72.2	D

The values with different capital letters (A–D) are significantly different (*p* < 0.05).

## Data Availability

All data created and analyzed during the experiments was presented in this study.

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
