# Peer review of "Polyphenols, L-Ascorbic Acid, and Antioxidant Activity in Wines from Rose Fruits (Rosa rugosa)"

_molecules, 2021, doi:10.3390/molecules26092561_

Round 1

Reviewer 1 Report

Principal weakness:

Reading the methods and materials section it looks like the trial was performed without biological replicates (and just with analytical replicates). This is a very big weakness to publish the results in a high impact factor journal such as Molecules. Obviously this fact is my main concern regarding the manuscript.

INTRODUCTION SECTION:

Given that fruit wines made from Rosa are little known beverage it is mandatory to include in the introduction section how these fruit wines are usually made (mixing the pulp or must with water, sugar addition, etc.).

RESULTS SECTION:

It is mandatory to include the alcoholic degree and sugar content of finished wines, in order to allow readers to understand the main features of the elaborated fruit wines from Rosa.

It is more appropriate to use mg/L than mg/100 mL for the concentrations of phenolics and ascorbic acid.

Line 123: How could be accidentally introduced such chelating metal ions?

Lines 123-125: Did Rosa contain high amounts of tartaric acid as happen with grapes?

Lines 179-180: It has not sense, since the source of antioxidant compounds are the raw material (rosa fruit). It is mandatory to include a reference if you want to mantain this affirmation in the text.

Lines 180-182: A reference about the antioxidant capacity of pectins is mandatory.

Lines 184-185: "increasing the content of polyphenols": it has no sense... Include a reference supporting this idea.

Lines 185-186: Include a reference supporting the sentence "Phenolic compounds can also be partially oxidized, which increases their antioxidant capacity"

Lines 186-187: Include a reference supporting the sentence "Esters of phenolic acids also show better antioxidant properties than phenolic acids alone"

Lines 187-189: Include a reference supporting that epigallocatechin gallate is a stronger antioxidant than gallic acid or catechins.

MATERIALS AND METHODS

Why did the authors use 50 °C during the enzymatic treatment? There are pectolytic enzymes working at lower temperatures... Such high temperatures could contribute to antioxidants loose.

The SO2 addition should to be indicated as the final content of SO2, instead of the added g of a 10% solution.

Author Response

Principal weakness

Reading the methods and materials section it looks like the trial was performed without biologlcal replicates (and just with analytical replicates). This is very big weakness to publish the results in a high impact factor journal such as Molecules. Obviously this fact is my main concem regarding the manuscript”.

The sample of Rosa Rugosa fruit (20 kg) from an industrial plantation was used for the study. The experiment was carried out in two technological duplicates (pages 12, 13 lines 353, 378).

INTRODUCTION SECTION

Given  that wines made from Rosa are little known beverage it is mandatory to include in the introduction section how these fruit wines are usually made (mixing the pulp or must with water sugar addition etc.”

Pages 1, 2; lines 42-81: additional information on wine production has been added to the introduction

RESULTS SECTION

-„It is mandatory to include the alcoholic degree and sugar content of finished wines in order to allow to understand the main features of the elaborated fruit wines from Rosa”

Physicochemical characteristics of the tested wines have been added (pages 2, 3, 4 and 13; lines 94-134, 380-392).

-„ It is more appropriate to mg/L. than mg/ 100 ml for the concentrations of phenolics and ascorbic acid”

In fact, the content of polyphenols and L-ascorbic acid in wines is usually expressed in g / L, but the authors also quote in mg / 100 ml. In the next works we will express the content in g / L.

- „Line 123 How could be accidentally introduced such chelating metal ions”

This sentence has been removed from the manuscript.

-“lines 123-125. Did Rose contain high amounts of tartaric acid as happen with grapes?”

The content of tartaric acid in rose fruits used for wine production was not determined. In the tested wines, only the total acid content was determined as malic acid (page 3, line 115-120 and page 13, line 387-391.

-“ Lines 179-180 It has not sense simple the source of antioxydant compound are the raw material (rosa fruit] It is mandatory to include a reference if you want to mantain this affirmation in the text „ Lines 180-182 A reference about the antioxidant capacity of pectin; is mandatory

Lines 184-185 ”increasing the content of polyphenols" it has no sense…. Include a reference supporting this idea”

„Lines 185-186- Include a reference supporting the sentence "Phenolic compounds can also be partially oxidized which increases their antioxidant capacity”

Lines 186-187 Include a reference supporting the sentence "Esters of phenolic acids also show better antioxidant properties than phenolic acids alone

Lines 187-189 Include a reference supporting that epigallocatechin gallate is a stronger antioxidant than than galic acid or catechines”

The literature confirming the statements used in the work is included (page 10, line 296-306)

MATERIALS AND METODS

 -„Why  did the authors use 500C during the enzymatic treatment? There are pectolytic enzymes working at lower temperatures. Such high temperatures could contribute to antioxidantes loose”.

The enzymatic treatment was carried out at the temperature of 50 ° C in accordance with the manufacturer's recommendation, Rohapect® 10 L. (page 12, line 366,367).

Czyżowska et al. also used enzymatic treatment at 50 degrees for 2 hours during the production of rose wines (Czyżowska, A.; Klewicka, E.; Pogorzelski, E.; Nowak, A. Polyphenols, vitamin C and antioxidant activity in wines from Rosa canina L. and Rosa rugosaThunb. Journal of Food Composition and Analysis 2015, 39, 62-68).

The S02 addition should to be indicated as the final content of S02 instead of the added g of a

10% solution

Completed in the Appendix A (page 16, line 491-493)

Reviewer 2 Report

Fermented alcoholic beverages are mainly divided into three types: traditional grape wines, beers and fruit wines. The latter use fruit other than grapes as basic ingredients, may contain additional aromas extracted from fruit, flowers or herbs and have the advantage of being rich in antioxidants. In this work the influence of the winemaking process on the antioxidant potential and on the content of phenolic compounds and L-ascorbic acid in wines from fruits of Rosa rugosa was evaluated.

General comments

The manuscript shows interesting aspects, in fact, although the economic, social, and cultural importance of the classic grape wine is recognized, nevertheless fruit wines can offer excellent flavors, colors and aromas that are very close to that of wines based on grapes.

Generally speaking, the paper is quite innovative (although there is a similar work from 2015 that the authors cite in the text, reference 24), the experimental plan is quite detailed and many experiments have been performed. However, while the part of the results is very detailed, the discussion is a bit poor and above all there is no comparison with the literature data. The ascorbic acid values ​​are in line with the literature values, while the total polyphenol values ​​are higher. On the other hand, as is known, the level of polyphenols also depends on factors such as agronomic practices, climatic conditions, soil orientation etc.

Overall I give a positive assessment to the work, however some changes need to be made to the text to make it suitable for publication. Therefore, I invite the authors to integrate the part of the discussion.

Minor points

Lines 77, 138, 178 and 336: Why in the text the authors write rose hips instead of Rosa rugosa?

References: The literature needs to be updated some articles are too old.

Author Response

“Overall I give a positive assessment to the work however sot-ne changes need to be made to the text to it suitable for publications Therefore I invite the authors to integrate the part of the discussion”

The discussion has been corrected (page 2-12)

-“Lines 77 138 178 and 33. Why in the text the authorswrite rose hips instead of Rosa  rugosa

The wording of rose hips in the sentences was changed to Rosa rugosa

“References. The literature needs to te updated some articles are too old

The literature was updated

Reviewer 3 Report

The manuscript entitled: „Polyphenols, L-ascorbic acid, and Antioxidant Activity in Wines from Rose Fruits (Rosa rugosa)” complements the information and provides novelity for science, future research, and industry. Research is properly planned and the work is concise, concrete and well structured. There are no objections to the results, their discussions and conclusions. Due to the specificity of the tests for the determination of antioxidant activity, the reviewer expresses doubts/ concerns that the use of only one test (DPPH) may not provide sufficient results to draw conclusions about the antioxidant activity.

Author Response

“Due to the specificity of the tests for the determination of antioxidant activity the reviewer expresses doubts/concerns that the use of only one test (DPPH) may not provide sufficient results to draw conclusions about the antioxidant activity”

The Folin–Ciocalteu method is usually used for determination of the total polyphenol content, but the reagent is nonspecific. The results of the determinations made by this  method are affected by the presence of reducing sugars, aromatic amines, sulphur dioxide, ascorbic acid, organic acids and other compounds, making the results often overstated. According to Prior and others and Huang and others [36,37] this method is based on oxidation–reduction reactions (single electron transfer-SET), and can thus be considered as one of the methods for the determination of antioxidant activity. This is also confirmed by the studies of other authors who observed a high regression coefficient between the amounts of substances reacted with the Folin- Ciocalteu reagent and antioxidant activity determined by DPPH test [38,39] (page 4, line 136-145).